# A Novel Transformer-Based IMU Self-Calibration Approach through On-Board RGB Camera for UAV Flight Stabilization

**DOI:** 10.3390/s23052655

**Published:** 2023-02-28

**Authors:** Danilo Avola, Luigi Cinque, Gian Luca Foresti, Romeo Lanzino, Marco Raoul Marini, Alessio Mecca, Francesco Scarcello

**Affiliations:** 1Department of Computer Science, Sapienza University, Via Salaria 113, 00198 Rome, Italy; 2Department of Mathematics, Computer Science and Physics, University of Udine, Via delle Scienze 206, 33100 Udine, Italy; 3Department of Computer Engineering, Modeling, Electronics and Systems Engineering, University of Calabria, Via Pietro Bucci, 87036 Rende, Italy

**Keywords:** UAV, deep learning, transformer, IMU, IMU calibration, computer vision

## Abstract

During flight, unmanned aerial vehicles (UAVs) need several sensors to follow a predefined path and reach a specific destination. To this aim, they generally exploit an inertial measurement unit (IMU) for pose estimation. Usually, in the UAV context, an IMU entails a three-axis accelerometer and a three-axis gyroscope. However, as happens for many physical devices, they can present some misalignment between the real value and the registered one. These systematic or occasional errors can derive from different sources and could be related to the sensor itself or to external noise due to the place where it is located. Hardware calibration requires special equipment, which is not always available. In any case, even if possible, it can be used to solve the physical problem and sometimes requires removing the sensor from its location, which is not always feasible. At the same time, solving the problem of external noise usually requires software procedures. Moreover, as reported in the literature, even two IMUs from the same brand and the same production chain could produce different measurements under identical conditions. This paper proposes a soft calibration procedure to reduce the misalignment created by systematic errors and noise based on the grayscale or RGB camera built-in on the drone. Based on the transformer neural network architecture trained in a supervised learning fashion on pairs of short videos shot by the UAV’s camera and the correspondent UAV measurements, the strategy does not require any special equipment. It is easily reproducible and could be used to increase the trajectory accuracy of the UAV during the flight.

## 1. Introduction

Nowadays, navigation systems play a key role in different scientific and industrial application contexts, including robotics [1] and autonomous vehicles [2]. However, the recently obtained results in this field show remarkable improvements and precision levels. The localization and the pose estimation of a device can be achieved with multiple strategies and techniques [3,4,5]; anyway, the choice of the involved sensors remains a crucial aspect. For instance, the global positioning system (GPS) [6] is one of the most commonly used systems for localizing and tracking devices. Concurrently, the inertial measurement unit (IMU) [7] can also be exploited due to its adaptive nature: it is accurate enough to identify a person [8], but it can even detect actions [9]. Moreover, the combination of multiple sensors [10] can often offer a noticeable improvement in terms of accuracy and precision. In the specific case of unmanned aerial vehicles (UAVs) [11], numerous tasks can be accomplished thanks to technological advancement, e.g., change detection [12,13], anomaly detection [14,15], and tracking [16,17]. However, there are still some open issues. Some of these may concern incorrect values obtained from the UAV sensors, especially when the device is not expensive. According to [18], for short paths, it is possible to ignore the drift caused by a low-cost IMU. It means that the error is feasible in real-life applications. However, it is only valid when the distance to cover is short; in fact, for long distances, the error propagation can be highly relevant, implying an unmanageable drift in applications where accurate precision is critical. In this context, machine learning (ML) and deep learning (DL) approaches are also feeding the research [19,20]. In the past years, the scholars exploiting transformer-based neural networks [21] caused a shockwave in the DL community due to their simplicity and state-of-the-art performances. Transformers are employed in a wide variety of domains: in natural language processing [22,23,24], computer vision [25,26,27], audio processing [28], and UAV applications [29,30,31]. Although the transformer-based networks are currently the standard in many tasks, their downside lies in the energy drain [32], the required computational power, and the needed primary memory, which is quadratic in the number of inputs for attention-based transformers [21].

## 2. Related Work

To the best of our knowledge, none of the works in literature treats the analyzed task with the same approach proposed in this paper. However, numerous similar applications inspired the presented strategy for a camera-based IMU self-calibration for UAVs.

One of the most relevant works considered in our analysis is [33], where the authors proposed an online IMU self-calibration method for visual–inertial systems equipped with a low-cost inertial sensor. It is specifically designed for unmanned ground vehicles (UGV), but it involves both IMU and camera sensors. It aims to locate the device as accurately as possible. In the first step, the IMU measurements and the camera frames are mapped. Since the image frequency is lower than the IMU updates, the IMU measurements between the two frames are pre-integrated into a single compound measurement. An optimization of the IMU intrinsic parameters is proposed to minimize the errors constrained by the pre-integration measurements. It is achieved by computing a Jacobian matrix [34] for each pre-integration part and estimating the uncertainty of the pre-integration measurements. The final step consists in defining the IMU intrinsic parameters residual. The authors propose an approach in which they are assumed to be uncertain, but which is constant in the time span of the sliding window. The proposed system can be classified as a monocular visual–inertial system (VINS-Mono) [35]. The results obtained on several datasets [36,37,38,39,40] show outstanding results in terms of tracking accuracy and also computational times.

IMU/camera calibration strategies can also be applied in other application areas [41] than those entailing UGVs or UAVs. It suggests that this approach can be considered a valid procedure, even if it seems uncommon. In the specific field of UAVs, it is possible to find different techniques in the recent literature, from the fusion of sensors [42] to the self-calibration [43,44]. For instance, in [45], the authors propose a novel method for monitoring a UAV with a custom-made positioning module that exploits a fusion algorithm. The work explains how the module should be calibrated to reduce the influence of deterministic errors on the IMU. The authors highlight five variables that are correlated to the noise factors on accelerometers and gyroscopes: direct measures, reference excitation acceleration, statistic bias, scale factor, and non-orthogonality. Based on these assumptions, the proposed method exploits a turnable platform to make a specific accelerometer’s axis point to gravity to obtain an output excitation, and it also rotates at a constant angular rate on a specific gyroscope’s axis for the same aim. After the measured output and reference excitation have been acquired, a least-squares method calculates the deterministic errors parameter. Similarly, the deterministic errors of the magnetometer are influenced by multiple factors. The correlated variables are the measured output, the matrix that contains errors like soft iron coefficients and non-orthogonality, the vector of static bias and hard iron coefficients, and the environment magnetic field distribution. The authors noticed how those parameters could provide an equation of a classical vector form of an ellipsoid formula; with an ellipsoid fitting, the geometry parameter can be acquired to calibrate the sensor. Concerning stochastic error identification, three noises have been highlighted: white noise, exponentially correlated noise, and rate random walk. The authors treated them by exploiting the Allan variance method and its accuracy approximation. Then, with system equations, the sixteen states, including attitude, velocity, position, and angle/velocity increment bias, are managed. A Kalman filter-based technique is introduced, in which measures and predictions support each other. The provided steps run to a fusion algorithm that combines all the sensors’ data with the extended Kalman filter approach and provides a more accurate real-time attitude, velocity, and position of the UAV.

Another example of multi-sensor calibration for UAVs is provided in [46], where the authors propose a correction method that involves different GPS sensors, a LIDAR, and an IMU. In the pipeline, three branches are shown: bore sight, time delay, and level arm calibrations. The first one aims to know exactly the relative position of sensors (multiple GPS, IMU, and LIDAR) with respect to each other. The authors provide a method for generating the coordinate transform of these sensors with respect to the center of the drone. Concerning the bore sight calibration, it is used to determine the differences in the rotations of each sensor. The proposed algorithm provides the alignment of the extracted and pre-processed point cloud data from the LIDAR, as well as a comparison with the GPS signals (that are considered the ground truth). It is also specified that the calibration could be even more precise if the IMU is also involved. The time delay calibration deals with the calculation of error-corrected signals. In particular, it refers to the delay in GPS sending and receiving messages. The solution consists in always adding the retrieved error to the location shown by the GPS receiver.

There are also works [47,48,49,50] in which the IMU’s calibration does not require any correlation with other sensors. As expected, most of the recent studies involve machine and deep learning-based techniques. For instance, in [51], the authors propose a calibration method based on deep learning for micro electro mechanical system (MEMS) IMU gyroscopes. The network calibrates the error from the raw data of the MEMS IMU and regresses gyroscope data after the error compensation. During the training, the network output is used to estimate the quaternion and calculate the loss function. During the test phase, the obtained output is exploited for navigation to obtain the altitude and position of the carrier. The results show that the position and the altitude are significantly adjusted by the proposed calibration method, providing relevant scientific soundness of the self-calibration approaches in this application field. Another example is provided by [52], where the introduction of an adaptive neuro-fuzzy inference system (ANFIS) is proposed to improve the effectiveness of low-grade IMUs by estimating and compensating the measurement errors. The method combines the artificial neural network (ANN) with the fuzzy inference system (FIS) and is trained by adding noise to reference IMU data. The proposed solution shows a high correction impact on the considered dataset, highlighting the quality of the strategy.

Concerning the specific deep learning technique of transformers, there are only a few works in the literature featuring this architecture in a UAV-based calibration task. The networks proposed in [53], one based on a long–short term memory (LSTM) architecture [54] and another on a transformer [21], estimating the bias in the sensors involved in the visual–inertial odometry in case of complete absence of visual information. Moreover, ref. [55] proposes a transformer-based architecture to pre-process and clean the IMU measurements before even being used in a flight. These cases suggested that the transformers seem promising in this application area and could provide noticeable improvements when correctly exploited.

## 3. Proposed Method

The model predicts the offset in the IMU of the drone given an input sequence $F\in R^{n\times h\times w}$ of *n* frames with height *h*, width *w*, and a sequence of aligned IMU measurements $I\in R^{n\times 6}$, containing records of accelerometer and gyroscope, each along the three axes. The first issue involves synchronization, which can be achieved by exploiting the IMU and the camera timestamps. However, it is worth considering that IMUs have a much higher sampling rate with respect to cameras and that these timestamps could not be perfectly matched. To address this issue, the practical solution is to match each camera frame with the closest corresponding IMU measurement in time. This also has some side effects. On one hand, it reduces the number of inputs for the model, thereby reducing the computational demand. On the other hand, this also ensures that IMU measurements are sampled at the same rate as cameras. This is a practical advantage since it makes the system scalable with the camera quality. To handle the multi-modal nature of the task, the model entails three blocks: a video reducer block (VRB) (Section 3.1) that aims to reduce the input sequence of consecutive frames into an equally long sequence of token vectors; an IMU reducer block (IRB) (Section 3.2) that projects each IMU vector, containing records of accelerometer and gyroscope in each instant of time, into a sequence of token vectors; and a noise predictor block (NPB) (Section 3.3), which merges the information from video and IMU records and predicts the amount of offset in the latter. Figure 1 depicts the model’s architecture. The source code of the presented strategy can be found at https://github.com/rom42pla/rgb2imu_error_estimation.

### 3.1. Video Reducer Block

This module, whose architecture is shown in Figure 1 and is partially inspired by [26,27], takes as input a sequence of *n* frames from a video and transforms each frame into a *d*-dimensional vector, called token, that can be merged with those returned by the IRB for the final prediction. In this context, the video can be grayscale, with $F\in R^{n\times h\times w}$ or RGB, with $F\in R^{n\times 3\times h\times w}$. The model is compatible with any combination of *n*, *h*, and *w*.

If a frame *F* is RGB, it is cast to a single channel via a small convolutional neural network (CNN) $CCN_{\chi }:R^{n\times 3\times h\times w}\to R^{n\times h\times w}$, including a three-dimensional convolution parameterized by χ, an activation function σ and a batch normalization layer [56]:(1)$$CNN_{\chi }\left(F\right)=batchnorm\left(\sigma \left(conv3d_{\chi }\left(F\right)\right)\right).$$

To ensure the preservation of the original information of each image, the output of the CNN is summed with the grayscale version of *F* as it is usually executed in residual networks [57]:(2)$$F=\left\{\begin{array}{cc} F & F\;\mathrm{is}\;\sin\mathrm{gle}\;\mathrm{channel},\\ grayscale\left(F\right)+CNN_{\chi }\left(F\right) & F\;\mathrm{is}\;\mathrm{RGB}. \end{array}\right.$$

Consequently, each frame is unfolded into a series of *p* smaller patches of shape $h_{p}\times w_{p}$, yielding a matrix $P\in R^{n\times p\times h_{p}\times w_{p}}$. If *h* is not divisible for $h_{p}$, or *w* is not divisible for $w_{p}$, the frames will be 0-padded (In this context, the image padding consists in adding extra pixels to perform an operation that requires a specific image shape. In a 0-padded image, the pixels added by the padding operation have an intensity of 0 on each channel.) until they fit the legal size for the unfolding operation. Each patch is then flattened into a $\left(h_{p}\;\cdot \;w_{p}\right)$-dimensional vector to be projected by a feed-forward network (A feed-forward network is often made of a sequence of one or more linear layers and activation functions. Moreover, also, other layer types, such as dropout [58] and normalization [56,59], can often be added, as well, depending on the context.) (FFN) into a *d*-dimensional token. This FFN, called $FFN_{\Theta }:R^{p\times (h_{p}\;\cdot \;w_{p})}\to R^{p\times d}$, entails a linear layer, also called fully connected, parameterized by a weights matrix $W_{\Theta }\in R^{d\times (h_{p}\;\cdot \;w_{p})}$ and a bias vector $b_{\Theta }\in R^{d}$. There are also an activation function σ and a dropout layer [58] for regularization purposes:(3)$$T_{p}=FFN_{\Theta }\left(P\right)=dropout\left(\sigma \left(flatten\left(P\right)W_{\Theta }^{T}+b_{\Theta }\right)\right).$$

After applying $FFN_{\Theta }$ individually to each of the *n* frames, the result will be a sequence of tokens $T_{p}\in R^{n\times p\times d}$.

The token of each patch is then provided to a transformer encoder to exploit the spatial dependencies between the patches. For each frame, it outputs a single token bearing the meaningful learned features towards the task. The transformer encoder architecture is analogous to the one originally presented in [21]. This module is made of a stack of one or multiple transformer encoder layers, which contain a sequence of multi-head attention (MHA) and an FFN with skip-connections. A skip-connection technique consists in using the output of a layer as input for another one placed later in the architecture. This method is commonly exploited in very deep networks to mitigate the vanishing gradient phenomenon [60]. The output of both the MHA and the FFN is further processed by a normalization layer [56,59]. A dot-product attention function takes three inputs $Q\in R^{n_{q}\times d_{q}}$, $K\in R^{n_{v}\times d_{k}}$, $V\in R^{n_{v}\times d_{v}}$, called query, key, and value, and is parameterized by three weights matrix $W_{q}\in R^{d_{q}\times d}$, $W_{k}\in R^{d_{k}\times d}$, and $W_{v}\in R^{d_{v}\times d}$:(4)$$Attention\left(Q,K,V\right)=softmax\left(\frac{QW_{q}{\left(KW_{k}\right)}^{T}}{\sqrt{d_{k}}}\right)VW_{v},$$
where $W_{q}$, $W_{k}$, and $W_{v}$ are used to transform the three inputs *Q*, *K*, *V* into a single output of shape $R^{n_{q}\times d}$. Therefore, the output of the attention function is a weighted sum of the values *V* based on the similarity between the query *Q* and the keys *K*. A MHA is a projection of the concatenation of the outputs of *k* dot-product attention operations, parameterized by different weights. The concatenation of the outputs of the attention operations is projected into a smaller space by a learned weight matrix $W_{o}\in R^{d\;\cdot \;k\times d}$:(5)$$MHA\left(Q,K,V\right)=\left(\overset{k}{\underset{i=1}{⨁}}Attention_{i}\left(Q,K,V\right)\right)W_{o}.$$

In a transformer encoder, the inputs of the attention functions are all equal to the input matrix, thus Q=K=V. In the VRB, there are multiple instances of the same transformer encoder $TE_{\Lambda }:R^{p\times d}\to R^{p\times d}$. Each one is fed with the patches of a single frame summed with $E_{\Lambda }\in R^{p\times d}$, a matrix of the learned positional embeddings. This is required since the attention Formula (Equation 4) does not model positional information [61] and cannot distinguish between patches in different positions. The output of a transformer encoder is a matrix of the same shape as the input, from which the model extracts the first row where all the meaningful information of the frame is squeezed, leading to a sequence of tokens $T_{p}^{\left(\mathrm{enc}\right)}\in R^{n\times d}$:(6)$$T_{p}^{\left(\mathrm{enc}\right)}=TE_{\Lambda }\left(T_{p}+E_{\Gamma }\right),$$
where, $T_{p}^{\left(\mathrm{enc}\right)}$ is further summed with the learned temporal embeddings $E_{\Gamma }\in R^{n\times d}$ and fed to a transformer encoder $TE_{\Gamma }:R^{n\times d}\to R^{n\times d}$ to transform $T_{p}^{\left(\mathrm{enc}\right)}$ into the output $T_{v}^{\left(\mathrm{enc}\right)}\in R^{n\times d}$ of the VRB. This allows analyzing the temporal relationships between tokens:(7)$$T_{v}^{\left(\mathrm{enc}\right)}=TE_{\Gamma }\left(T_{p}^{\left(\mathrm{enc}\right)}+E_{\Gamma }\right).$$

### 3.2. IMU Reducer Block

This module, whose architecture is shown in Figure 1, takes as input a sequence of *n* IMU measurements $I\in R^{n\times 6}$ and transforms each of them into a *d*-dimensional vector, called a token. Each measurement is time-aligned with the corresponding video frames. Although this paper considers 6 typologies of values in the formulations, the model is compatible with any combination of standard IMU measurements to accommodate any device. For instance, if records from both the accelerometer and gyroscope are present in *I*, then its last dimension will be 6, since three values are for the axes of the accelerometer and three for the gyroscope. Meanwhile, as in the case of the experiments on the Zurich MAV dataset [39], if the measurements of only the axes of one of the two sensors are available, *I* will contain n×3 values.

The measurements *I* are processed by $FFN_{\Xi }:R^{n\times 6}\to R^{n\times d}$, entailing two linear layers, respectively, parameterized by $W_{\Xi ,1}$, $b_{\Xi ,1}$ and $W_{\Xi ,2}$, $b_{\Xi ,2}$, to project these vectors into the same space of the remaining tokens in the model:(8)$$T_{i}=FFN_{\Xi }\left(I\right)=dropout\left(dropout\left(\sigma \left(IW_{\Xi ,1}^{T}+b_{\Xi ,1}\right)\right)W_{\Xi ,2}^{T}+b_{\Xi ,2}\right).$$

Similarly to the VRB, the projected tokens are summed to the learned positional embeddings $E_{\Psi }\in R^{n\times d}$ and fed to a transformer encoder [21] $TE_{\Psi }:R^{n\times d}\to R^{n\times d}$, parameterized by Ψ, to inject the temporal information into the inputs:(9)$$T_{i}^{\left(\mathrm{enc}\right)}=TE_{\Psi }\left(T_{i}+E_{\Psi }\right).$$

The model’s output is a sequence of tokens $T_{i}^{\left(\mathrm{enc}\right)}\in R^{n\times d}$ that bear the learned features from the IMU measurements for an analyzed video.

### 3.3. Noise Predictor Block

This module, whose architecture is shown in Figure 1, is used to merge the tokens of both VRB and IRB. To this aim, a transformer encoder parameterized by Φ$TE_{\Phi }:R^{n\times d}\to R^{n\times d}$, is employed:(10)$$T^{\left(\mathrm{enc}\right)}=TE_{\Phi }\left(T_{v}^{\left(\mathrm{enc}\right)}+T_{i}^{\left(\mathrm{enc}\right)}+E_{\Phi }\right).$$

As for the other transformer encoders in the model, $E_{\Phi }\in R^{n\times d}$ are the learned positional embeddings used to inform the module about the temporal relationships between input tokens. Then, a sequence of a transformer decoder and a FFN is used to make the final predictions. In this context, a transformer decoder [21] is similar to a transformer encoder, entailing a stack of one or more transformer decoder layers. Differently from transformer encoders, it takes two matrices as input, called target and memory. A transformer decoder layer is made of a sequence of two multi-head attentions and an FFN with skip connections. Although the first MHA takes the target matrix as input for the *Q*, *K*, and *V* parameters, the second one takes the target as *Q* and the memory as *K* and *V*, effectively injecting the target matrix with information about the target. In this model, the memory is equal to the output of $TE_{\Phi }$, representing information about the input frames and the IMU measurements. Instead, the target is represented by a learned matrix $L\in R^{1\times d}$ that represents the embedding of the query of the final offsets vector in a latent *d*-dimensional space. This transformer decoder $TD_{\Upsilon }:R^{1\times d}\times R^{n\times d}\to R^{1\times d}$, parameterized by Υ, outputs the embedding of the offsets vector that will be then projected into a 6-dimensional space by $FFN_{\Pi }:R^{d}\to R^{6}$, which contains a single linear layer parameterized by $W_{\Pi }\in R^{6\times d}$ and $b_{\Pi }\in R^{6}$: (11)$$\hat{o}=TD_{\Upsilon }\left(L,T^{\left(\mathrm{enc}\right)}\right)W_{\Pi }^{T}+b_{\Pi },$$
where $\hat{o}$ will contain the predicted offsets of the IMU of the drone.

### 3.4. Training Strategy

The calibration problem is solved using a supervised learning paradigm. In this context, the system is trained to approximate a function given a training dataset composed of labeled samples. The learned function estimates the label of the unknown samples, minimizing a prediction error called loss [62]. In order to be processed by the model, the inputs are segmented into fixed duration windows, where each represents a sample of the training $D_{t}$ or the validation set $D_{v}$. Each sample is a tuple $\left(F,I\right)$ made of a sequence of *n* grayscale ($F\in R^{n\times h\times w}$) or RGB ($F\in R^{n\times 3\times h\times w}$) frames and their corresponding time-aligned IMU measurements $I\in R^{n\times 6}$. The goal of the model is to estimate the amount of noise in the input signal to correct the misalignment of the sensor. To this aim, we combine the input signal with a generated noise that must be plausible in a real-world IMU. Let $o\in R^{6}$ be a vector representing the noise that, in this context, corresponds to the offsets of the IMU. This vector is randomly sampled from a Gaussian distribution $N(\mu ,n_{m}\sigma^{2})$, with $\mu ,\sigma \in R^{6}$ and $n_{m}\in R$. The latter is a noise multiplier that has been set to 2 in the experiments. The elements $\mu_{i}$ and $\sigma_{i}$ for each i∈[1,6] are the mean and standard deviation of the *i-th* elements of all the IMU measurements in the dataset. In other words, each position $o_{i}$ with i∈[1,6] is randomly filled with a value taken into the symmetric interval $\left[\mu_{i}-n_{m}\sigma_{i},\mu_{i}+n_{m}\sigma_{i}\right]$. At the training time, the noisy IMU measurements, $I^{\left(\mathrm{noisy}\right)}$, are computed as follows:(12)$$I^{\left(\mathrm{noisy}\right)}={\left[I_{a}+o\right]}_{a=1}^{n},$$
where *n* is the number of frames and IMU measurements in the sample (F,I). During the training, a different o is generated for each sample to increase data variability and, consequently, robustness. During the validation, instead, o is generated once and applied to all the samples in the test. This choice guarantees the fairness of the results since, in real-world scenarios, when an IMU is miscalibrated, this displacement is constant.

The model *M*, described in Section 3 and parameterized by Ω, is then fed with the frames and the noisy IMU values and aims to predict an estimation of the offsets. It is trained to minimize a loss function L between the predicted and the ground truth offsets:(13)$$\min_{\Omega }L\left(\hat{o},o\right),$$
where, $\hat{o}$ is a shorthand for $M_{\Omega }\left(F,I^{\left(\mathrm{noisy}\right)}\right)$. The optimizer of the proposed model is AdamW [63], a modified version of Adam [64], including decoupled weight decay. It iteratively adjusts the model’s weights via gradient descent to minimize Equation (Equation 13). The loss function L used to train the network is the sum of three terms obtained using the mean squared error (MSE) between the predicted offsets $\hat{o}\in R^{6}$ and the ground truth offsets $o\in R^{6}$. The mean squared error function is defined in Equation (Equation 14) and returns a real number given two input vectors of the same shape:(14)$$MSE\left(\hat{y},y\right)=\frac{1}{n}\sum_{i=1}^{n}{\left({\hat{y}}_{i}-y_{i}\right)}^{2}.$$

The objective is to output an approximation of the noise that minimizes the dissimilarity between o and $\hat{o}$, and, thus, the first term is the MSE between them:(15)$$L_{o}\left(\hat{y},y\right)=MSE\left(\hat{y},y\right).$$

However, the use of the $L_{o}$ loss made the model learn values near the mean of the distribution of the offsets, whose value is around 0. Two more auxiliary losses have been used to enforce the model to predict more meaningful values. $L_{m}$ enforces the modulus, also called magnitude, of each element in $\hat{o}$ to be as near as possible to the magnitude of each corresponding element in o:(16)$$L_{m}\left(\hat{y},y\right)=MSE\left(|\hat{y}\left|,|y\right|\right).$$

This penalizes the model stagnation around low values when the ground truth is high, solving one of the problems. To enforce the correctness of the sign of each element in $\hat{o}$, the term $L_{s}$ is used:(17)$$L_{s}\left(\hat{y},y\right)=MSE\left(sign\left(\hat{y}\right),sign\left(y\right)\right).$$

Usually, if the sign of a value is wrongly predicted, the error is directly proportional to the expected magnitude. Thanks to $L_{s}$, which can be seen as an uncertainty indicator, the model can automatically regulate itself in case of uncertainty and be less bold in predicting high magnitudes. The final loss function is defined as follows:(18)$$L\left(\hat{y},y\right)=L_{o}\left(\hat{y},y\right)+L_{m}\left(\hat{y},y\right)+L_{s}\left(\hat{y},y\right).$$

## 4. Experiments and Discussion

### 4.1. Datasets

The proposed strategy was tested on three datasets during the experiments to demonstrate its effectiveness. This section aims to highlight their main characteristics.

The EuRoC Micro Aerial Vehicle dataset (https://projects.asl.ethz.ch/datasets/doku.php?id=kmavvisualinertialdatasets), here referred to as EuRoC MAV [37], contains data from 11 indoor experiments in an industrial environment. The videos are available at a resolution of 752×480 pixels with a single channel. Besides the onboard sensors’ records, the dataset comes with a point-cloud representation of the scenarios obtained from external LIDAR sensors.

The University of Zurich First-Person View dataset (https://fpv.ifi.uzh.ch/datasets/), here referred to as UZH-FPV [65], is a collection of 28 flight experiments recorded with a grayscale camera in different environments and with different setups for the drone. The scenarios are indoor and outdoor, and the camera is pointed either forward or at a 45° angle downwards. Each video has been recorded at a resolution of 346×260 pixels, while the IMU data are composed of accelerometer and gyroscope records.

The Zurich Urban Micro Aerial Vehicle dataset (https://rpg.ifi.uzh.ch/zurichmavdataset.html), here referred to as Zurich MAV [39], is composed of a single, long record of the states of a drone while flying in an urban scenario. The flight covers around 2 Km at an altitude varying from 5 to 15 m. The provided data includes records from the on-board IMU, the Google Street View snapshots of the path, and the frontal camera acquisitions. In particular, the video has been shot at a resolution of 1920×1080 pixels and with three RGB channels.

### 4.2. Metrics

This section introduces the metrics used in the result evaluation provided in Section 4.3 to quantify the model’s error when estimating the offsets in the IMU. The root mean squared error (RMSE) is a common metric to quantify the error in the regression tasks. Given a vector of predicted values $\hat{y}\in R^{n}$ and a vector of ground-truth values of the same shape $y\in R^{n}$, and this error is computed as the squared root of the mean squared error (described in Equation (Equation 14)):(19)$$\begin{array}{cc} RMSE(\hat{y},y) & =\sqrt{MSE(\hat{y},y)}=\\ \; & =\sqrt{\frac{1}{n}\sum_{i=1}^{n}{\left({\hat{y}}_{i}-y_{i}\right)}^{2}}. \end{array}$$
RMSE is usually preferred to the MSE since its scale is the same as the input vectors instead of its square. In addition, it also supports humans to better visualize and understand the value. The $F_{1}$ score is a standard metric to quantify the error in the classification task. This measure is a particular case of the more general $F_{\beta }$ measure that expresses the classification error in terms of two other metrics called precision and recall:(20)$$precision\left(TP,FP\right)=\frac{TP}{TP+FP},$$
(21)$$recall\left(TP,FN\right)=\frac{TP}{TP+FN}.$$
where, TP, FP, and FN are the True Positives, False Positives, and False Negatives, respectively. They are calculated by comparing a vector of predicted values $\hat{y}\in L^{n}$ and a vector of correct values $y\in L^{n}$, where *L* is the set of all possible labels. The classification task in this paper is binary, meaning that there are two possible classes for each element: positive and negative signs, leading to |L|=2. In this setting, a TP represents an element that is correctly classified; a FP, also called a Type 1 error, represents a negative element wrongly classified as positive; and a FN, also called a Type 2 error, represents a positive element wrongly classified as negative. However, precision does not take into account the FN, while recall does not consider the FP. Given that the two measures are complementary, $F_{\beta }$ is a way to consider them both and is defined as: (22)$$\begin{array}{cc} F_{\beta }\left(TP,FP,FN\right) & =\frac{(1+\beta^{2})\;\cdot \;precision\;\cdot \;recall}{\beta^{2}\;\cdot \;precision+recall}=\\ \; & =\frac{(1+\beta^{2})\;\cdot \;TP}{\left(1+\beta^{2}\right)\;\cdot \;TP+\beta^{2}\;\cdot \;FN+FP}, \end{array}$$
with β∈R>0 denoting the impact of the recall with respect to the precision. In this paper, the β=1 version of $F_{\beta }$ is used, called $F_{1}$ score. This value of β is a common procedure in many classification tasks and leads to equal consideration of the two measures:(23)$$\begin{array}{cc} F_{1}\left(TP,FP,FN\right) & =\frac{2\;\cdot \;precision\;\cdot \;recall}{precision+recall}=\\ \; & =\frac{2\;\cdot \;TP}{2\;\cdot \;TP+FN+FP}. \end{array}$$

### 4.3. Results and Discussion

A trial is a syntactic sugar word that consists of a full training and validation procedure. Two kinds of validation schemes [66] have been adopted for the experiments. The *k*-fold cross-validation scheme has been used in the trials on the Zurich MAV dataset since it is composed of a single video/experiment. In this scheme, windows/samples are partitioned into *k* groups called folds. To this aim, *k* different trials are performed using a different fold each time as the validation set and the rest as the training set, averaging the results between each training. The Leave-One-Out (LOO) cross-validation schema, used in UZH-FPV and EuRoC MAV, is a particular case of *k*-fold cross-validation. In this context, for each trial, the validation fold is each time made of samples from a single video/experiment and the rest of the dataset as the training set. The total number of trials reported is 49:11 for EuRoC MAV, 28 for UZH-FPV, and 10 for Zurich MAV. In the tables, it is worth pointing out that the most relevant parameter is the offset since it represents the overall displacement between the IMU’s measurements and the real values in a real context scenario. The chosen size for the windows/samples is 10 s since, as shown in Table 1, longer sizes yield better results. Increasing the dimension of data provided to the architecture could probably allow for extracting more meaningful information from each sample. However, going too far with the windows’ size could negatively impact the results since it will reduce the number of available samples. At the same time, it will also significantly increase the amount of memory required for training, limiting the proposal’s applicability.

The final configuration of the model has been inferred from the results of the ablation study (Section 4.4) and is the same for each trial and dataset. Table 2 presents a complete overview of the parameters. It is worth noticing that some parameters described in the table are not optimal according to the ablation study in Section 4.4, such as the number of attention heads and the size of the patches. The presented configuration is the one providing the best trade-off between regression and classification performance, as well as computation speed. The following lines describe in more detail the most meaningful ones. The dropout amount has been set to 1% since it has been empirically found that bigger dropouts are one of the major causes of performance drops among the model’s parameters. The hidden size (referred to as *d* in Section 3) has been set to 256 since it provides the best tradeoff between the $F_{1}$ on signs prediction and the RMSE on magnitudes and offsets. The number of encoder and decoder layers in each transformer encoder and decoder has been set to 2, while the attention heads are 8. The noise multiplier $n_{m}$, as already mentioned in Section 3.4, is the scalar factor used with the standard deviation of the noise distribution to generate the offsets o. It has been empirically set to 2, since it covers reasonable IMU offsets. The size of the patches (referred to as $h_{p}\times w_{p}$ in Section 3) has been set to 64×64 since it works much better than a size of 32×32 and slightly worse than 96×96. This results in a good compromise between the two sizes in terms of performance since bigger sizes require more epochs to converge, as shown in Section 4.4.

The starting learning rate of the optimizer has been set to $10^{-5}$ as in other state-of-the-art works, although AdamW is adaptive, and thus this parameter is not so impactful on the final results. Additionally, videos are first resized to a resolution of 320×180 pixels; this operation is performed to have a fixed h=180 and w=320, but it also enables stacking multiple videos into batches. The experiment’s timings and the hardware configuration are shown in Table 3.

The model validation results can be seen in Table 4 and graphically in Figure 2, where sample predictions are shown for six samples of each dataset.

It is interesting to notice that the model is more firm when predicting the offsets on samples from EuRoC MAV and Zurich MAV, whose videos are taken in similar scenarios, while it is more uncertain when dealing with samples from UZH-FPV. In fact, the latter is the only dataset that contains both indoor and outdoor environments, as well as different camera angles (forward-facing and 45° downward-facing). Another possible reason may be the lower resolution of the images: 346×260 pixels, compared to the 752×480 pixels of EuRoC MAV and the 1920×1080 pixels of Zurich MAV. Although the proposed strategy re-scales all the frames to 320×180 pixels as a pre-processing step, low-quality starting images may hinder the results.

### 4.4. Ablation Study

The ablation study consisted of a grid search of the optimal value of several hyper-parameters. There were a series of trials on the UZH-FPV dataset where each one is performed using the base configuration of the model, described in Section 4.3 and Table 2, with a single variation of a parameter. LOO and *k*-fold cross-validation schemes would require training several instances of the same model on different splits of the dataset and averaging the results. However, from a computational point of view, this would be infeasible. Given that, we chose the compromise to keep fixed training and validation sets to only train a single model once for each combination of parameters. The training and the validation sets are, respectively, random 80% and 20% of windows in the dataset. This is a common proportion in the related works and is inspired by the Pareto principle [67]. The results of the ablation study are shown in Table 5 and graphically in Figure 3. Each row in Table 5 represents the trial’s results using different values from the base configuration for specific parameters. An early stopping mechanism has been implemented to stop a trial whenever the training loss does not improve for three consecutive epochs. The metrics used are the same as Section 4.3: the $F_{1}$ on signs and RMSE on offsets and magnitudes. The parameters that have been tested are the amount of dropout, the hidden size *d*, the number of attention heads in the MHA, the number of encoders or decoders in each transformer encoder and transformer decoder, the multiplier for the noise added to o, and the dimension of the patches in the VRB $h_{p}\times w_{p}$.

One of the most impactful parameters is the amount of dropout, which yields a drop of 0.236 in the RMSE offsets considering a dropout of 1% or 20%. As can be noticed from the table, for this kind of task, increasing the percentage of dropped connections highly worsens the results. A smaller hidden size *d* reduces the overfitting but considerably stretches the training time, as can be seen in Figure 3. A hidden size equal to 512, which corresponds to the highest one tested, performs 1.54% better on the $F_{1}$ of the signs with respect to a hidden size of 256. However, the latter value improves the RMSE of offsets and magnitude, respectively, by 0.021 and 0.008 points. Models with 4 and 8 attention heads have been tested. The lowest value outperforms the other by 1.05% on the $F_{1}$ of the signs and nearly 0.02 on the RMSE of offsets and magnitude. Having more attention heads also implies a higher computation demand. However, as it happens for the hidden size and number of layers parameters, a higher number of attention heads lead to a decrease in convergence time. Increasing the number of layers also decreases the convergence time as for Figure 3. On average, the improvement of a model with 4 layers is 0.067 on the RMSE of the offsets and 0.051 on the RMSE of the magnitudes. Considering the $F_{1}$ on the signs, 4 is not the best value. However, on average, it improves by 2.32%, and it is only 0.13% worse than the best result achieved by a model with 2 layers. A higher noise multiplier $n_{m}$ hinders the capability of the model to accurately distinguish between real signal and noise, as shown in Figure 3. The model performs slightly better on the two RMSEs if the multiplier is set to ×2 since these values represent more plausible noises. Since the estimation of the sign is a binary classification task, it seems that increasing $n_{m}$ helps distinguish the two classes. In fact, the gap between a $n_{m}$ of ×4 and ×1 is 7.69 percentage points on the $F_{1}$ signs metric. The size of the patches, inversely proportional to the number of tokens in the first transformer encoder in the VRB, seems to perform better if set with higher values, as can be seen in Figure 3. The worst performing size resulted in 32×32, which is 16.88% worse on the $F_{1}$ signs with respect to a size of 96×96, 0.43 worse on the RMSE offsets and 0.222 worse on the RMSE magnitude. Anyway, higher patches severely mine the computational times and memory requirements. For this reason, we chose the 64×64 since it would make a good compromise for a real application on a drone.

## 5. Conclusions

This work presents a novel neural network architecture for camera-based IMU calibration for UAVs. These procedures are crucial in contexts where the UAV navigation could be only based on a predetermined trajectory to reach the final destination (e.g., in war fields where the GPS signals are usually hijacked) and, in particular, when the route to cover is long. The contribution of this proposal is three-fold. In the first place, it does not make any use of special equipment or impractical hardware procedures to correct the offsets in the sensors, as it could be necessary for other kinds of calibration strategies. In the second place, the proposed pre-trained model can be used directly on the UAV, making it available in a wider range of scenarios. In third place, the algorithm is compatible with both grayscale and RGB images, accommodating different cameras mounted or integrated into diverse UAV models. Moreover, it does not require normalizing the IMU measurements in input, minimizing the computational efforts. More in detail, the algorithm is a transformer-based neural network that processes both a window of sequential frames recorded from the on-board camera and the IMU measurements to estimate the eventual offsets of the latter. The model comprises three modules called video reducer block, IMU reducer block, and noise predictor block. The first two reduce the video and the sequence of the IMU measurements to sequences of tokens of the same shape. These are then summed and given to the latter block to make the final offsets prediction. The approach has been tested on three datasets that are commonly used in the literature for other tasks. Two of them (EuRoC MAV, UZH-FPV) have low-resolution, grayscale-only videos, while the other (Zurich MAV) has full-HD RGB images. Interestingly, the method does not behave much differently on grayscale or RGB images, making it almost camera-agnostic. As expected, the uncertainties are fewer in the two more static datasets and slightly higher in UZH-FPV, which features different environments and camera angles.

One of the issues lies in the fact that the token mixing parts of the transformer encoders and decoders are based on the standard attention mechanism, in which complexity scales quadratically with the number of inputs (e.g., the number of frames of the video and the number of patches in each image). Although this strategy can be considered the standard in transformer-based works, we plan to replace the token, mixing parts with attention-free mechanisms to alleviate the time and space requirements of the neural network. It is worth reminding that, since the offsets are usually small, an IMU calibration has a tangible impact mostly for long traversals. For this reason, one effective way to evaluate the proposed approach would be to analyze data collected from a UAV flying along a straight path for a considerable distance (e.g., 1 Km) and compare the displacement between the predicted and actual final positions. However, to the best of our knowledge, no existing dataset contains data from UAV flights that meet these criteria. We plan to validate the proposed procedure in real-flight test scenarios, in which it is required that the UAV follows a predetermined trajectory toward a destination.

## Figures and Tables

**Figure 1 sensors-23-02655-f001:**
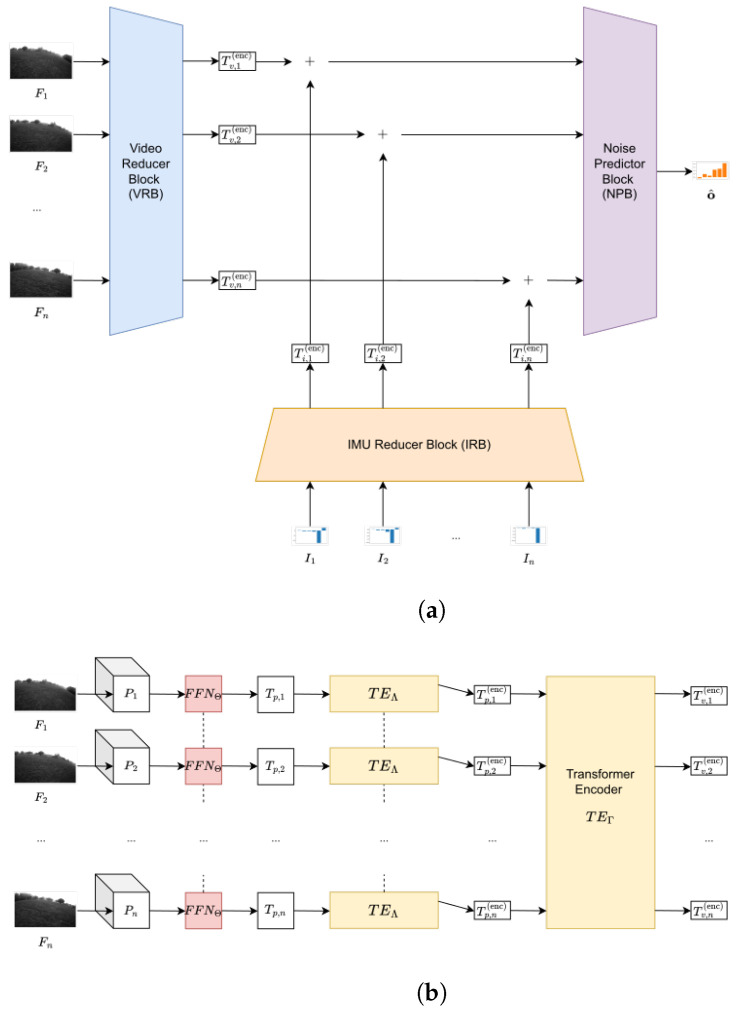
The architecture of the model. A dashed line between two modules means weight sharing between them. (**a**) Overview of the full model. The outputs of the video reducer block (VRB) and of the IMU reducer block (IRB) are summed and passed to the noise predictor block that will output the final offset estimation $\hat{o}\in R^{6}$; (**b**) (overview of the video reducer block (VRB)). The input frames $F\in R^{n\times h\times w}$ are unfolded into patches that are projected into token vectors and processed by two transformer encoder to output a token for each frame $T_{v}^{\left(\mathrm{enc}\right)}\in R^{n\times d}$; (**c**) (overview of the IMU reducer block (IRB)). The input IMU measurements $I\in R^{n\times 6}$ are projected into a *d*-dimensional space and then processed by a transformer encoder to output a representation $T_{i}^{\left(\mathrm{enc}\right)}\in R^{n\times d}$ of the measurements; (**d**) Overview of the noise predictor block (NPB). The tokens $T\in R^{n\times d}$ obtained from the sum of the outputs of the VRB and the IRB are passed through a transformer encoder and its results $T^{\left(\mathrm{enc}\right)}\in R^{n\times d}$ to a transformer decoder along with a query matrix $L\in R^{1\times d}$. The output is the predicted offsets $\hat{o}\in R^{6}$.

**Figure 2 sensors-23-02655-f002:**
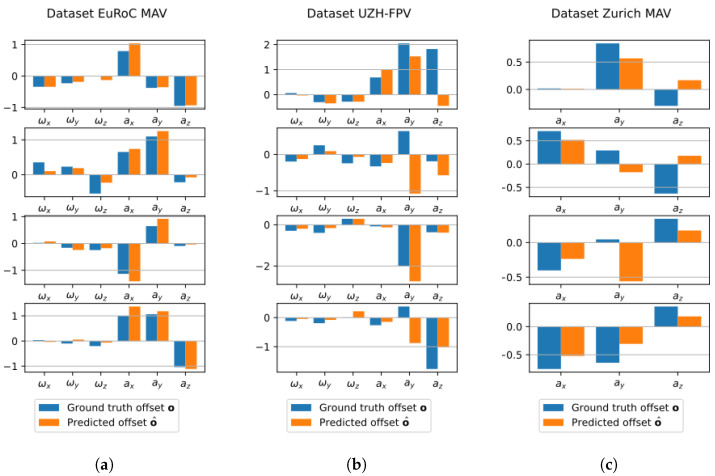
Sample predictions of the model on the datasets. Values $\omega_{x}$, $\omega_{y}$, and $\omega_{z}$ are, respectively, the measurement of the gyroscope for the three axes, while $a_{x}$, $a_{y}$, and $a_{z}$ are the corresponding ones from the accelerometer. (**a**) EuRoC MAV [37]; (**b**) UZH-FPV [65]; (**c**) Zurich MAV [39].

**Figure 3 sensors-23-02655-f003:**
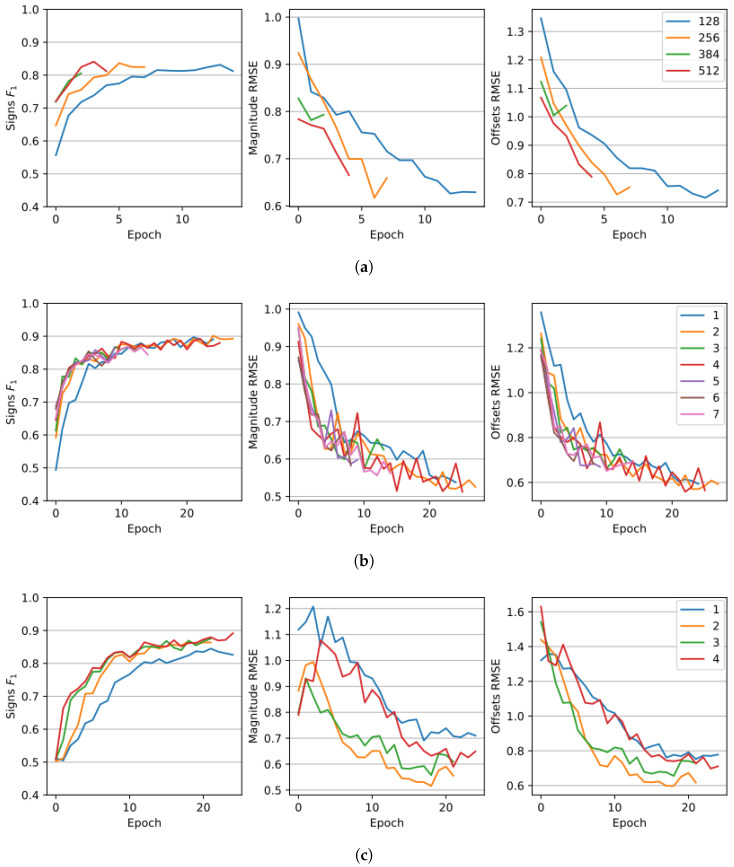
Plots of the results of the ablation study on four parameters of the model. (**a**) Hidden size (*d*). (**b**) Number of layers in the encoders and the decoders. (**c**) Injected noise multiplier ($n_{m}$). (**d**) Size of the patches ($h_{p}$, $w_{p}$).

**Table 1 sensors-23-02655-t001:** Results of the experiments at various window sizes. The experiments have been performed on UZH-FPV [65] dataset, the most challenging and various among the ones analyzed in this work. Train and validation sets follow an 80/20% split, fixed between experiments and chosen as described in Section 4.4. The stride of the windows is kept fixed to 1 s alongside the seed. The best value in each result column is in bold.

Window Size	Signs $F_{1}$	Offsets RMSE
2 s	87.4%	0.686
4 s	87.9%	0.616
6 s	87.4%	0.532
8 s	90.1%	0.563
10 s	**91.8**%	**0.495**

**Table 2 sensors-23-02655-t002:** The configuration of the model used in the final evaluations.

Parameter	Value
Dropout	1%
Hidden size (*d*)	256
Attention heads	8
Layers	2
Noise multiplier ($n_{m}$)	2×
Patches size ($h_{p}\times w_{p}$)	64×64
Batch size	$\left\{\begin{array}{cc} 16 & \mathrm{dataset}\;\mathrm{is}\;\mathrm{EuRoC}\;\mathrm{MAV}\;\mathrm{or}\;\mathrm{UZH}-\mathrm{FPV}\\ 8 & \mathrm{dataset}\;\mathrm{is}\;\mathrm{Zurich}\;\mathrm{MAV} \end{array}\right.$
Activation function (σ)	Rectified Linear Unit: ReLU(x)=max(0,x)
Learning rate	$10^{-5}$

**Table 3 sensors-23-02655-t003:** Times and stats of the experiments. All the experiments have been run sequentially on a cluster’s node with 64 GB DDR4 RAM memory, 1× AMD Epyc 7301 2.2
GHz CPU core, and a Nvidia Quadro 6000 GPU with 24 GB of RAM.

**Dataset**	**Validation Type**	**Number of Trials**	**Batch Size**
EuRoC MAV [37]	LOO	11	16
UZH-FPV [65]	LOO	28	16
Zurich MAV [39]	10-fold	10	8
**Dataset**	**Epochs per Trial**	**Time per Trial**	**Time per Every Trial**
EuRoC MAV [37]	17.5	803.2 s	8835.4 s
UZH-FPV [65]	13.6	1136.3 s	31,815.4 s
Zurich MAV [39]	4.7	1214.3 s	12,142.9 s
**Dataset**	**Batches per Training Epoch**	**Time per Training Step**	**Time per Training Epoch**
EuRoC MAV [37]	69	0.607 s	41.9 s
UZH-FPV [65]	133	0.620 s	82.5 s
Zurich MAV [39]	300	0.801 s	240.2 s
**Dataset**	**Batches per Validation Epoch**	**Time per Validation Step**	**Time per Validation Epoch**
EuRoC MAV [37]	10	0.412 s	4.1 s
UZH-FPV [65]	3	0.422 s	1.3 s
Zurich MAV [39]	33	0.551 s	18.2 s

**Table 4 sensors-23-02655-t004:** Results on the three datasets. Each numerical cell is composed of the mean and standard deviation between the best values of each validation split.

Dataset	Validation	Signs $F_{1}$	Magnitude RMSE	Offsets RMSE
EuRoC MAV [37]	LOO	94.59%/1.55%	0.177/0.044	0.193/0.062
UZH-FPV [65]	LOO	89.61%/3.57%	0.454/0.164	0.502/0.203
Zurich MAV [39]	10-fold	73.79%/3.84%	0.273/0.038	0.360/0.049

**Table 5 sensors-23-02655-t005:** Results of the ablation study. Variations of several parameters of the model have been tested, and this table shows the results of the best score on the validation set of each trial. The best value in each result column for a particular parameter is in bold.

Parameter	Value	Signs $F_{1}$	Magnitude RMSE	Offsets RMSE
Dropout	0%	**80.62%**	**0.631**	0.750
Dropout	1%	80.53%	0.635	**0.749**
Dropout	5%	78.29%	0.662	0.792
Dropout	10%	72.30%	0.741	0.932
Dropout	20%	70.51%	0.776	0.986
Hidden size	128	78.51%	**0.720**	**0.862**
Hidden size	256	79.67%	0.733	**0.862**
Hidden size	384	79.39%	0.788	1.022
Hidden size	512	**81.21%**	0.728	0.883
Attention heads	4	**79.04%**	**0.663**	**0.793**
Attention heads	8	77.99%	0.684	0.818
Layers	1	83.51%	0.669	0.780
Layers	2	**85.79%**	0.610	0.706
Layers	3	83.82%	0.661	0.789
Layers	4	85.66%	**0.602**	**0.699**
Layers	5	82.17%	0.667	0.786
Layers	6	81.55%	0.673	0.778
Layers	7	83.22%	0.637	0.755
Noise multiplier	1×	74.83%	0.879	0.968
Noise multiplier	2×	78.49%	**0.665**	**0.809**
Noise multiplier	3×	80.76%	0.690	0.839
Noise multiplier	4×	**82.52%**	0.800	0.946
Patches size	32×32	59.97%	0.919	1.276
Patches size	64×64	73.89%	0.750	0.929
Patches size	96×96	**76.85%**	**0.697**	**0.841**
Patches size	128×128	72.04%	0.787	0.976

## Data Availability

Not applicable.

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
