# Peer review of "A Novel Transformer-Based IMU Self-Calibration Approach through On-Board RGB Camera for UAV Flight Stabilization"

_sensors, 2023, doi:10.3390/s23052655_

Round 1

Reviewer 1 Report

In this study, a soft calibration procedure based on Transformer Neural Network is proposed. The study is scientifically well designed and explained. Acceptable after minor revision.

1) I don't think it is necessary to capitalize the first letters of expressions like Transformer-based Neural Networks, Computer Vision.

2) How was 80% and 20% training-test data determined for UZH-FPV? Information on this subject can be given.

3) Discussion can be added for the results in Table 5 in the Ablation study section.

Author Response

Dear Reviewer,

please, see the attachment.

Best regards,

Marco Raoul Marini

Reviewer 2 Report

I read the article with great interest, because the problem posed in it is extremely relevant and the solution promised by the authors is a lifeline/panacea for all designers of navigation system of unmanned autonomous vehicles. The first two chapters left a very good impression. Unfortunately, the method presented in the third chapter and the obtained results in the experimental chapter do not convince me that the proposed soft calibration method works and can lead to reducing the misalignment in inertial sensors' readings.

Author Response

(The authors gave the same response as above.)

Reviewer 3 Report

My comments regards to this paper as the following:

o is a vector representing the noise, please describe more about this parameter, how its distribution, mean and std...

- Attention head (in the Table 5) indicated that 4 is better than 8, but in the Table 2 show the value of attention head is 8. Please explain

- In the Eq. 18, Lo maybe significant lower than Lm and Ls, so please consider add contribution weights between them (Lo, Lm, and Ls).

Attention heads

Author Response

(The authors gave the same response as above.)

Round 2

Reviewer 2 Report

I want to thank the authors for their kind invitation to visit their GitHub repository to reproduce the experiments on the selected datasets and different ones to check the correctness of the results and the method's effectiveness for the calibration task. But despite my desire to do so, no such GitHub page exists. Please do not invitate the reviewers, because there are people who will respond to them and find nothing!

However, I am inclined to allow the article to be published with some remarks to it.

3.1. Video Reducer Block. On page 4,6 transformer encoder is described and in equations (4) and (5) the three inputs Q, K, V are used without any description. The attention function also needs an additional description.

3.2. IMU Reducer Block. It is assumed that each IMU measurement is time-aligned with the corresponding video frames. In real life this is not the case and special measures should be provided for the time synchronization of IMU measurements with camera frames. It should be stated even if it is not the subject of research.

Page 7 Two inaccuracies were made in the description of the Gaussian noise parameters. The first of these is that 99.7% of random realizations of Gaussian noise with parameters \mu and \sigma fall in the interval \mu +- 3\sigma or the mentioned symmetric interval will be [\mui-3nm\sigmai, \mui-3nm\sigmai]

The second inaccuracy is in the description of the noise vector, which in the given description assumes the same offset relative to the mathematical expectation.

Finally, I will conclude with a recommendation. The aim of the article is the calibration of the inertial sensors based on the images obtained from the camera. Some quantitative measures of the work done are presented in Fig. 2. The measures used, however, do not allow potential users to assess what the real benefit will be from the application of the proposed new approach in a UAV control. The user is much more interested in the error in orientation and position, by how much it has decreased.

Author Response

Dear Reviewer,

You can find the answers in the attached letter.

Best regards,

Marco
